# Gender differences in anxiety and perceived stress during the Covid-19 pandemic among Brazilian adolescents from impoverished communities

**Nicolas Kimura Generoso**[1], **Ana Luiza Vilela Borges**[2], **Cristiane da Silva Cabral**[1]*

**1** Department of Health and Society, School of Public Health, University of São Paulo, São Paulo, Brazil,
**2** Department of Public Health Nursing, School of Nursing, University of São Paulo, São Paulo, Brazil

* cabralcs@usp.br

## Abstract

This study examines factors associated with anxiety and perceived stress among adolescents living in socioeconomically vulnerable urban areas of São Paulo, Brazil, in the context of the Covid-19 pandemic. Drawing from a cross-sectional analysis of 396 cisgender adolescents aged 13–17, recruited through Family Health Strategy units, data were collected via structured home-based interviews and analyzed using validated psychometric tools (GAD-7 and PSS-10). Findings indicate that 28.5% of participants exhibited moderate to severe anxiety, and 72.5% reported moderate to high levels of perceived stress, especially among girls. Multivariate models identified age, gender, perceived emotional neglect, experiences of verbal hostility within the household, digital sociability, and pandemic-induced academic difficulties as key predictors of adverse outcomes. Conversely, protective associations were observed for adolescents whose parents were in a relationship, who reported affective support at school, or who lived in owned housing. The intersection between gender norms, emotional vulnerability, and structural deprivation emerged as a critical lens for interpreting the psychosocial consequences of the pandemic. These results underscore the need for context-sensitive public health strategies that transcend individualizing frameworks and address the material, relational, and symbolic dimensions of adolescent mental health in unequal urban settings.

## Background

The global pandemic of the Coronavirus Disease 2019 (Covid-19) has left an indelible mark on various domains, including humanitarian, economic, and political realms, with consequences that are yet to be fully quantified [1,2]. On children and adolescents, this scenario exerted a detrimental influence on the overall physical and psychological well-being [3].

**Data availability statement:** The anonymized dataset that supports the findings of this study is available at http://fsp.usp.br/geas_brasil/data.csv. The database has also been submitted as Supporting Information.

**Funding:** This publication was supported by a subagreement from Johns Hopkins University with funding provided by Grant No. 135844 from AstraZeneca UK (to CSC), administered through the Charities Aid Foundation (CAF), and by the São Paulo Research Foundation (FAPESP), under Grant No. 2017/23177-4 (to ALVB) and Grant No. 2024/15093-9 (to NKG). The content is solely the responsibility of the authors and does not necessarily represent the official views of AstraZeneca UK, the Charities Aid Foundation, Johns Hopkins University, or FAPESP. The funders played an important role only in developing the instrument; study design, data collection and analysis, decision to publish, or preparation of the manuscript had no participation of the funders.

**Competing interests:** The authors have declared that no competing interests exist.

In Brazil, the immunization campaign against Severe Acute Respiratory Syndrome Coronavirus 2 (SARS-CoV-2), the etiologic agent causing Covid-19, commenced in March 2021, approximately one year after the onset of the pandemic. This development instilled a ground for optimism and hope within the Brazilian population, as it constituted a noteworthy triumph in the fight against the virus [4]. However, it was not until September of the same year that the campaign was expanded to encompass adolescents between the ages of 12 and 17. It was thus determined that adolescents with comorbidities should be considered a priority group for vaccination, to be followed by the general population of the same age group, who are the participants in this study.

As they progress through their developmental years, which are shaped by changes in their relational, identity, and subjective experiences, young people are confronted with an intensification of these complex circumstances [5]. In addition to the typical challenges associated with their age, adolescents also had to contend with the inherent adversities of their realities, which were further compounded by the necessity of coexisting with the virus [6]. This has resulted in an even greater vulnerability, given that this demographic is more sensitive to stress and anxiety due to the combination of uncertainties and concerns about the SARS-CoV-2 infection and the measures taken to contain it, such as physical distancing and home confinement [7–9].

Approximately 90% of school-aged youth have been impacted physically, mentally, emotionally, and educationally on a global scale [10]. Brazilian adolescents endured nearly two years of social isolation due to school closures, a longer period than that experienced by adolescents in other middle- and high-income countries [11]. The restrictions on social interaction with individuals outside the family nexus have been identified as a contributing factor to difficulties in stress and anxiety management when confronted with novel circumstances and impediments to emotional expression [12].

Such difficulties exacted a notable toll on the mental health of the adolescents. The extant literature suggests that girls may be more vulnerable to unfavorable outcomes than their male counterparts with respect to both stress [13] and anxiety [14]. Nevertheless, shifts in the response to the pandemic by boys and girls may reflect internalized social gender patterns. The traditional socialization norms attributed to boys may inhibit the expression of emotional distress and limit their likelihood of seeking support, contributing to the underreporting of psychological suffering [15]. Within patriarchal and colonially structured contexts such as Brazil, however, girls are frequently subjected to more rigorous behavioral regulation and intensified domestic scrutiny, which may heighten their exposure to psychological forms of violence [16]. These facts may help elucidate the more frequent acknowledgment of anxiety and perceived stress among girls, who, in turn, tend to adopt more proactive stances in recognizing and articulating emotional discomfort [17,18].

Emerging evidence bring out the disproportionate impact of the Covid-19 pandemic on adolescent mental health in Brazil, where social vulnerability and structural inequality intensified emotional distress. Data from the 2004 Pelotas Birth Cohort,

with follow-up conducted between November 2019 and March 2020, revealed marked declines in adolescents' emotion regulation, self-esteem, and behavioral self-control, particularly among girls and those exposed to socioeconomic stressors [19]. Barros et al. (2022) [20], analyzing a national sample of 9,470 Brazilian adolescents, identified high levels of sadness (32.4%) and nervousness (48.7%) during social distancing, strongly associated with being female, older age, financial hardship, and household conflict. Similarly, Cabral et al. (2023) [8] conducted a qualitative study in São Paulo's peripheries, revealing how remote schooling, disrupted peer ties, and gendered domestic burdens amplified adolescents' emotional suffering, particularly among girls. These results collectively show that adolescent mental health during the pandemic was not merely an individual concern, but one embedded in broader matrices of inequality, precarious living conditions, and weakened support networks – reinforcing the need for context-sensitive analyses like the present study.

Data from the 2022 census indicate that adolescents aged 10–19 comprise approximately 14% of the Brazilian population over – 30 million individuals – of whom nearly 60% reside in urban territories marked by acute socioeconomic vulnerability, such as favelas and marginalized neighborhoods [21]. This study was conducted in São Paulo, the country's most populous metropolis, with over 12 million inhabitants, among whom an estimated 12% are adolescents. Notably, more than 1.7 million people inhabit the city's peripheral zones [21], which are emblematic of the enduring socio-spatial segregation that structures São Paulo's urban fabric. These peripheral regions disproportionately concentrate racialized populations, reduced access to public services, and chronic exposure to economic precarity, factors that collectively shape the psychosocial environments in which adolescents navigate their formative years.

This paper examines two pivotal elements of mental health: perceived stress and anxiety. The term "perceived stress" is employed to delineate an individual's subjective appraisal of the stressfulness of the circumstances they confront, and its subsequent impact on their physical and emotional well-being [22]. In the case of "anxiety", it is an emotional response involving feelings of tension, worriedness, and physiological alterations [23,24]. The study adds to the literature on the mental health consequences of the Covid-19 pandemic for adolescents in structurally vulnerable settings. Brazil, one of the countries that was most severely affected by the pandemic, recorded over 4,000 deaths per day in early 2021 [25], highlighting the urgency of the present investigation. We explored the associations between a range of factors (sociodemographic characteristics, family dynamics and structure, adversities at home, peer sociability and leisure activities, school setting and pandemic hurdles in education) and levels of anxiety and perceived stress among adolescents aged 13–17 in the post-pandemic period, with particular attention to gender-based standoffs.

## Methods

### Ethical clearance

The research proposal was approved by the Ethics Committee of the School of Public Health of the University of São Paulo (protocol number 568742). São Paulo Municipal Health Department provided the requisite authorization for the fieldwork. The consent form was signed by the parents of the adolescents, while the assent form was signed by the adolescents themselves. An option to decline to answer any question was made available. Furthermore, adolescents were permitted to terminate their participation at any point and complete the survey in multiple sessions. No financial incentive was provided to any adolescent for their participation in the study.

### Study sample

This is a cross-sectional study that falls under the umbrella of the Global Early Adolescent Study (GEAS), a Covid-19 module conducted in Brazil and other countries from the Americas, Africa, Europe, and Asia [5,26,27]. This study employed a non-probabilistic sample of 406 adolescents aged 13–17, residing in Itaim Paulista Administrative District, located in the eastern zone of São Paulo. Participants were identified through records from Family Health Strategy (FHS) facilities, accessed with authorization from the São Paulo Municipal Health Department.

PLOS Mental Health

## Data collection and Measures

Data collection took place between November and December 2021 through home visits conducted by trained health professionals, accompanied by Community Health Workers (CHWs) who were familiar with the local territory. Adolescents were invited to participate during these visits. Given the non-probabilistic nature of the sample, the findings are not generalizable to the broader adolescent population of São Paulo. This sampling approach may have introduced selection bias by overrepresenting households more strongly connected to primary health care services or more easily accessible to CHWs; however, it depicts well a poor neighborhood in the city.

Participants completed a structured, self-administered questionnaire on electronic tablets under the supervision of qualified health professionals. The instrument was used to assess dimensions of mental health in the period following the most critical phase of the pandemic. To this end, two widely recognized scales for measuring anxiety and perceived stress among adolescents were employed as our outcomes: the Generalized Anxiety Disorder 7-Item Scale "GAD-7" [23] and the Perceived Stress Scale 10-Item "PSS-10" [22]. The anonymized dataset is available as supporting information (S1 Data).

The GAD-7 was developed to assess levels of anxiety characterized by excessive, persistent, disproportionate, and difficult-to-control worriedness in various domains of life [23]. A recent study demonstrated its efficacy as a screening tool for anxiety in Brazilian adolescents [28]. The scale comprises seven items, with each item requesting that participants indicate the intensity of their symptoms over the previous two weeks on a Likert scale ranging from "Not at all" to "Almost every day". A score between 10 and 14 indicates moderate anxiety, while a score between 15 and 21 indicates severe anxiety. Consequently, adolescents with a score of 10 or above are deemed to exhibit a "moderate to severe level of anxiety". This cutoff (≥ 10) was originally proposed by Spitzer et al. (2006) [23], the authors of the scale, which showed excellent properties for identifying generalized anxiety disorder in adults in primary health care settings (sensitivity = 0.89; specificity = 0.82) [23]. Its applicability in the Brazilian context has been supported by subsequent studies. Capellini et al. (2023) [29], for example, adopted this threshold in a sample of Brazilian physiotherapists, based on a meta-analysis by Plummer et al. (2016) [30] that confirmed the robustness of the ≥ 10 cutoff across different populations and cultural settings.

The PSS-10, comprising ten items, was initially devised by Cohen et al. (1983) [22] to evaluate individuals' perception of the unpredictability, uncontrollability, and overload of life in the previous month, and how this influenced their emotional state. The objective was to reflect the degree to which everyday situations are perceived as stressful. The scale has been validated for use in Brazil among college professors [31]. The response options range from "Never" to "Almost always", with scores from 0 to 40 (0–13: low stress; 14–26: moderate stress; 27–40: high stress). For this study, adolescents with a score of 14 or above were classified as having a "moderate to high level of perceived stress". This cutoff (≥ 14) aligns with established conventions for the PSS-10 observed in Wiriyakijja et al. (2020) [32], who adopted this threshold in a clinical validation study. Although no culturally specific cutoff has been formally established for the Brazilian adolescents, our categorization does not contravene the interpretive boundaries of the scale. Rather, we opted to group the moderate and high stress categories to better capture more pronounced levels of distress, following methodological precedents grounded in the literature [22,31,32].

In both scales, responses of "I do not know", "I do not remember", "I refuse to answer", or "Other" were excluded to mitigate the potential for incomplete or ambiguous information to interfere with the results [33]. To gain insight into the elements that may be associated with mental health outcomes, the predictor variables were grouped into the following domains: 1) sociodemographic characteristics (gender; age; race/skin color; religious affiliation; school's administrative domain; owned housing; mother's schooling); 2) family dynamics and structure (socializing with siblings; parents or guardians work; adult parents or guardians are in a relationship; household headed exclusively by a woman); 3) adversities at home (perceived increase in the volume of domestic tasks; lacked sufficient financial resources to procure adequate nourishment; feeling afraid or very bad about being insulted/rejected by adults at home; feeling of lack of love and care from people at home; witnessing/experiencing situations of aggression, threats, or beatings at home); 4) peer sociability and leisure activities (spent time hanging out with friends in person; reached out to friends via messages or social networks;

chatted with friends via messages or social networks; used social networks to chat with friends or interacted with computer games or other media; TV use; perceived improvement in the amount of contact with friends during the pandemic; perceived improvement in the quality of relationships with friends during the pandemic); 5) school setting (level of schooling; level of schooling to be achieved; feeling cared for/protected at school by other adults) and 6) pandemic hurdles in education (the pandemic has had a negative effect on the ability to focus on classes; the pandemic has had a negative effect on the ability to get good grades at school; considered dropping out of school).

## Statistical analysis

Descriptive analyses were conducted using absolute and relative frequencies. To assess whether the proportions of girls and boys differed significantly across variables, Pearson's chi-square test was applied. Subsequently, binomial logistic regression was used to examine associations between explanatory variables within each analytical domain and the two outcomes of interest: moderate to severe anxiety and moderate to high perceived stress.

The strength of associations was expressed as crude and adjusted odds ratios (OR and ORadj), accompanied by 95% confidence intervals (CIs). Univariate logistic regression models were initially estimated to evaluate the independent association between each explanatory variable and the outcomes. Variables with $p < 0.05$ in the domain-specific multivariate analyses were retained for inclusion in the final multiple logistic regression model.

The final model incorporated all variables that: a) were statistically significant in the stratified multivariate models; b) were found to be confounders – i.e., changed 10% or more the odds ratio when compared crude and adjusted estimates; c) were retained based on theoretical relevance. Multicollinearity was assessed using the variance inflation factor (VIF), with variables presenting VIF ≥ 5 excluded from the final model. Model fit was evaluated through the Hosmer-Lemeshow goodness-of-fit test and Nagelkerke's pseudo $R^2$. All statistical analyses were conducted in RStudio (version 2023.06.1 + 524) [34].

## Results

A total of 406 adolescents were interviewed, 396 of whom identified as cisgender, which is the population about whom the results are presented. The female participation was slightly predominant (51.3%). Much of the sample was Black (63.1%), attended public schools (90.7%), and had parents who were employed (86.4%). A total of 21.5% of respondents indicated that they lacked sufficient financial resources to procure adequate nourishment during the pandemic. Adolescents reported feelings of fear due to being insulted/rejected or the absence of love at home at a rate of 34.6% and 37.7%, respectively. There was a statistically significant difference between boys and girls in these rates ($p = 0.001$). Most respondents (79.4%) utilized digital communication platforms, such as text messages and social networks, to maintain contact with friends during the pandemic, yet 31.3% engaged in in-person interactions with friends. The pandemic had no negative effect on the ability to focus on classes for 65.2% of adolescents, nor on their ability to get good grades (72.8%) (Table 1).

Table 1 illustrates that there were statistically significant differences between boys and girls with regard to religion affiliation ($p = 0.027$), the administrative domain of the school ($p = 0.019$), TV use ($p = 0.040$), perceived improvement in the quality of relationships with friends during the pandemic ($p = 0.044$), the level of schooling to be achieved ($p = 0.002$), and feelings of care/protection at school by other adults ($p = 0.041$). About the outcomes on mental health, nearly one-third of the participants (28.5%) were classified as having a moderate to severe level of anxiety according to the GAD-7 scale, with a higher proportion of girls ($p = 0.001$). Additionally, 72.5% were classified as having a moderate to high level of perceived stress according to the PSS-10 scale. This finding was also observed in a higher proportion of female participants ($p = 0.001$) (Table 1).

Boys were less likely to be classified at the moderate to severe level of anxiety compared to girls [$ORa_{dj} = 0.33$; 95%CI 0.13-0.79]. Greater odds of moderate to severe anxiety were observed among participants who reported feeling

PLOS Mental Health

**Table 1. Sample description according to variables of interest and their relationship between genders. São Paulo, 2021.**

| Variables | Total | Girls | Boys | |
|---|---|---|---|---|
| | n (%) | n (%) | n (%) | p-value |
| *Sociodemographic characteristics* | | | | |
| **Age** | | | | |
| < 14 | 117 (29.5) | 54 (26.6) | 63 (32.6) | 0.227 |
| ≥ 14 | 279 (70.5) | 149 (73.4) | 130 (67.4) | |
| **Race/skin color** | | | | |
| Non-Black | 146 (36.9) | 84 (41.4) | 62 (32.1) | 0.071 |
| Black | 250 (63.1) | 119 (58.6) | 131 (67.9) | |
| **With a religious affiliation** | | | | |
| No | 105 (26.5) | 64 (31.5) | 41 (21.2) | **0.027** |
| Yes | 291 (73.5) | 139 (68.5) | 152 (78.8) | |
| **School's administrative domain** | | | | |
| Public | 341 (90.7) | 183 (94.3) | 158 (86.8) | **0.019** |
| Private | 35 (9.3) | 11 (5.7) | 24 (13.2) | |
| **Owned housing** | | | | |
| No | 150 (39.3) | 74 (37.4) | 76 (41.3) | 0.495 |
| Yes | 232 (60.7) | 124 (62.6) | 108 (58.7) | |
| **Mother's schooling** | | | | |
| Up to high school diploma or technical degree | 287 (82.2) | 149 (82.8) | 138 (81.7) | 0.893 |
| Higher education or above (incomplete or complete) | 62 (17.8) | 31 (17.2) | 31 (18.3) | |
| *Family dynamics and structure* | | | | |
| **Socializing with siblings** | | | | |
| No | 110 (27.8) | 61 (30.0) | 49 (25.5) | 0.372 |
| Yes | 285 (72.2) | 142 (70.0) | 143 (74.5) | |
| **Parents or guardians work** | | | | |
| No | 41 (13.6) | 23 (14.1) | 18 (13.0) | 0.788 |
| Yes | 260 (86.4) | 140 (85.9) | 120 (87.0) | |
| **Adult parents or guardians are in a relationship** | | | | |
| No | 153 (42.1) | 87 (45.6) | 66 (38.4) | 0.166 |
| Yes | 210 (57.9) | 104 (54.4) | 106 (61.6) | |
| **Household headed exclusively by a woman** | | | | |
| No | 296 (74.9) | 149 (73.4) | 147 (76.6) | 0.468 |
| Yes | 99 (25.1) | 54 (26.6) | 45 (23.4) | |
| *Adversities at home* | | | | |
| **Perceived increase in the volume of domestic tasks** | | | | |
| No | 256 (71.9) | 137 (70.6) | 119 (73.5) | 0.552 |
| Yes | 100 (28.1) | 57 (29.4) | 43 (26.5) | |
| **Lacked sufficient financial resources to procure adequate nourishment[1]** | | | | |
| No | 284 (78.5) | 141 (75.4) | 143 (81.7) | 0.144 |
| Yes | 78 (21.5) | 46 (24.6) | 32 (18.3) | |
| **Feeling afraid or very bad about being insulted/rejected by adults at home** | | | | |
| No | 229 (65.4) | 96 (53.6) | 133 (77.8) | **<0.001** |
| Yes | 121 (34.6) | 83 (46.4) | 38 (22.2) | |
| **Feeling of lack of love and care from people at home** | | | | |
| No | 223 (62.3) | 92 (50.3) | 131 (74.9) | **<0.001** |
| Yes | 135 (37.7) | 91 (49.7) | 44 (25.1) | |

*(Continued)*

**Table 1.** (Continued)

| Variables | Total | Girls | Boys | |
|---|---|---|---|---|
| | n (%) | n (%) | n (%) | p-value |
| **Witnessing/experiencing situations of aggression, threats, or beatings at home** | | | | |
| No | 322 (88.0) | 168 (89.8) | 154 (86.0) | 0.263 |
| Yes | 44 (12.0) | 19 (10.2) | 25 (14.0) | |
| *Peer sociability and leisure activities* | | | | |
| **Spent time hanging out with friends in person** | | | | |
| No | 244 (68.7) | 133 (72.3) | 111 (64.9) | 0.166 |
| Yes | 111 (31.3) | 51 (27.7) | 60 (35.1) | |
| **Reached out to friends via messages or social networks** | | | | |
| No | 74 (20.6) | 37 (19.8) | 37 (21.4) | 0.806 |
| Yes | 286 (79.4) | 150 (80.2) | 136 (78.6) | |
| **Chatted with friends via messages or social networks** | | | | |
| No | 148 (41.3) | 74 (40.0) | 74 (42.8) | 0.594 |
| Yes | 210 (58.7) | 111 (60.0) | 99 (57.2) | |
| **Used social networks to chat with friends or interacted with computer games or other media[2]** | | | | |
| No | 162 (48.6) | 74 (45.1) | 88 (52.1) | 0.246 |
| Yes | 171 (51.4) | 90 (54.9) | 81 (47.9) | |
| **TV use[2]** | | | | |
| No | 263 (76.9) | 123 (71.9) | 140 (81.9) | **0.040** |
| Yes | 79 (23.1) | 48 (28.1) | 31 (18.1) | |
| **Perceived improvement in the amount of contact with friends during the pandemic** | | | | |
| No | 187 (55.5) | 105 (60.3) | 82 (50.3) | 0.081 |
| Yes | 150 (44.5) | 69 (39.7) | 81 (49.7) | |
| **Perceived improvement in the quality of relationships with friends during the pandemic** | | | | |
| No | 166 (48.7) | 94 (54.3) | 72 (42.9) | **0.044** |
| Yes | 175 (51.3) | 79 (45.7) | 96 (57.1) | |
| *School setting* | | | | |
| **Level of schooling** | | | | |
| Elementary school | 277 (72.5) | 141 (71.9) | 136 (73.1) | 0.885 |
| High school | 105 (27.5) | 55 (28.1) | 50 (26.9) | |
| **Level of schooling to be achieved** | | | | |
| High school | 85 (27.8) | 32 (20.0) | 53 (36.3) | **0.002** |
| Higher education | 221 (72.2) | 128 (80.0) | 93 (63.7) | |
| **Feeling cared for/protected at school by other adults** | | | | |
| No | 37 (11.4) | 25 (15.2) | 12 (7.5) | **0.041** |
| Yes | 288 (88.6) | 139 (84.8) | 149 (92.5) | |
| *Pandemic hurdles in education* | | | | |
| **The pandemic has had a negative effect on the ability to focus on classes** | | | | |
| No | 221 (65.2) | 108 (61.0) | 113 (69.7) | 0.091 |
| Yes | 118 (34.8) | 69 (39.0) | 49 (30.3) | |
| **The pandemic has had a negative effect on the ability to get good grades at school** | | | | |
| No | 246 (72.8) | 117 (68.4) | 129 (77.2) | 0.089 |
| Yes | 92 (27.2) | 54 (31.6) | 38 (22.8) | |
| **Considered dropping out of school** | | | | |
| No | 335 (89.3) | 174 (90.6) | 161 (88.0) | 0.507 |
| Yes | 40 (10.7) | 18 (9.4) | 22 (12.0) | |

*(Continued)*

**Table 1.** (Continued)

| Variables | Total | Girls | Boys | |
|---|---|---|---|---|
| | n (%) | n (%) | n (%) | p-value |
| *Outcome variables* | | | | |
| **Classified as having a moderate to severe level of anxiety (GAD-7)** | | | | |
| No | 248 (71.5) | 99 (56.3) | 139 (87.1) | **<0.001** |
| Yes | 99 (28.5) | 77 (43.7) | 22 (12.9) | |
| **Classified as having a moderate to high level of perceived stress (PSS-10)** | | | | |
| No | 91 (27.5) | 29 (17.2) | 62 (38.3) | **<0.001** |
| Yes | 240 (72.5) | 140 (82.8) | 100 (61.7) | |

[1]Referring to the preceding month when the survey was carried out.

[2]Use of more than 3 hours/day.

Values in bold indicate statistical significance (p<0.05).

afraid or very bad when insulted or rejected by adults at home [$OR_{adj}$=3.79; 95%CI 1.36-10.97], as well as among those who experienced a perceived lack of love and care within their household [$OR_{adj}$=3.06; 95%CI 1.10-8.63]. Those who reported the use of digital communication tools to remain connected with peers during the pandemic, such as text messaging, social media, online games, or other digital platforms, were more likely to report moderate to severe level of anxiety than those who did not [$OR_{adj}$=5.08; 95%CI 2.08-13.52]. A similar pattern was found among adolescents who reported that the pandemic negatively affected their academic performance [$OR_{adj}$=2.24; 95%CI 1.02-5.49]. Conversely, adolescents who had parents in a relationship were less likely to report moderate to severe level of anxiety than their counterparts [$OR_{adj}$=0.38; 95%CI 0.15-0.87]. All results refer to a model adjusted for age, exposure to domestic violence (witnessing or experiencing aggression, threats, or beatings at home), and level of schooling that the adolescents aimed to achieve (Table 2).

Additional factors associated with moderate to high levels of perceived stress among adolescents included gender, age, owned housing, feelings of lack of love at home, and feelings of care/protection at school by other adults. As observed in relation to the moderate to severe level of anxiety, gender was also a determining factor in the classification of the moderate to high level of perceived stress: boys were markedly less likely to be categorized in the moderate to high stress group when compared to girls [$OR_{adj}$=0.28; 95%CI 0.12-0.62], reinforcing the gendered nature of psychological vulnerability in this population. Age also showed a significant positive association [$OR_{adj}$=2.43; 95%CI 1.08-5.56], which suggests that older adolescents may be more susceptible to cumulative or anticipatory stressors, possibly linked to academic expectations, identity conflicts, or shifting family roles.

Housing status was inversely associated with perceived stress: adolescents living in homes owned by their families were significantly less likely to report moderate to high levels of stress compared to the adolescents who did not live in owned households [$OR_{adj}$=0.29; 95%CI 0.13-0.64]. This result points to the stabilizing effect of residential security and its potential buffering role against daily stressors, particularly in socioeconomically vulnerable settings. Emotional experiences within the household also played a critical role: adolescents who reported feeling unloved or uncared for by family members were substantially more likely to exhibit higher levels of stress [$OR_{adj}$=4.80; 95%CI 1.71-15.0] than adolescents who did not, highlighting the centrality of affective ties and emotional availability in the home environment for adolescent mental health.

Conversely, the perception of being cared for and protected by adults in the school context was identified as a protective factor [$OR_{adj}$=0.18; 95%CI 0.01-0.98]. This finding underscores the importance of school as a place not only for academic development but also for psychosocial support, particularly when other spheres of socialization, such as the family, may be marked by neglect or conflict. All associations were adjusted for school administrative structure, female-headed

**Table 2. Selected variables and their associations with a moderate to severe level of anxiety, as measured by the GAD-7. São Paulo, 2021.**

| Selected variables | n (%) | Moderate to severe level of anxiety (GAD-7) | | OR [95%CI] | p-value | Binary Logistic Models | | | |
|---|---|---|---|---|---|---|---|---|---|
| | | No | Yes | | | Model 1* | | Model 2** | |
| | | % | % | | | OR$_{adj}$ [95%CI] | p-value | OR$_{adj}$ [95%CI] | p-value |
| *Sociodemographic characteristics* | | | | | | | | | |
| **Gender** | | | | | | | | | |
| Girls | 203 (51.3) | 39.9 | 77.8 | 1.00 | **<0.001** | 1.00 | **<0.001** | 1.00 | **0.012** |
| Boys | 193 (48.7) | 60.1 | 22.2 | 0.19 [0.11-0.32] | | 0.21 [0.11-0.38] | | 0.33 [0.13-0.79] | |
| **Age** | | | | | | | | | |
| < 14 | 117 (29.5) | 31.0 | 25.3 | 1.00 | 0.280 | 1.00 | 0.656 | 1.00 | 0.247 |
| ≥ 14 | 279 (70.5) | 69.0 | 74.7 | 1.33 [0.79-2.29] | | 1.15 [0.63-2.16] | | 0.52 [0.17-1.56] | |
| **Race/skin color** | | | | | | | | | |
| Non-Black | 146 (36.9) | 36.3 | 35.4 | 1.00 | 0.869 | 1.00 | 0.956 | – | – |
| Black | 250 (63.1) | 63.7 | 64.6 | 1.04 [0.64-1.70] | | 0.98 [0.55-1.78] | | – | |
| **With a religious affiliation** | | | | | | | | | |
| No | 105 (26.5) | 25.8 | 33.3 | 1.00 | 0.162 | 1.00 | 0.758 | – | – |
| Yes | 291 (73.5) | 74.2 | 66.7 | 0.70 [0.42-1.16] | | 0.91 [0.50-1.69] | | – | |
| **School's administrative domain** | | | | | | | | | |
| Public | 341 (90.7) | 88.7 | 93.8 | 1.00 | 0.147 | 1.00 | 0.426 | – | – |
| Private | 35 (9.3) | 11.2 | 6.2 | 0.53 [0.19-1.24] | | 0.64 [0.19-1.84] | | – | |
| **Owned housing** | | | | | | | | | |
| No | 150 (39.3) | 38.4 | 44.8 | 1.00 | 0.283 | 1.00 | 0.207 | – | – |
| Yes | 232 (60.7) | 61.6 | 55.2 | 0.77 [0.48-1.23] | | 0.70 [0.39-1.22] | | – | |
| **Mother's schooling** | | | | | | | | | |
| Up to high school diploma or technical degree | 287 (82.2) | 80.0 | 81.4 | 1.00 | 0.780 | 1.00 | 0.766 | – | – |
| Higher education or above (incomplete or complete) | 62 (17.8) | 20.0 | 18.6 | 0.91 [0.47-1.69] | | 1.12 [0.54-2.26] | | – | |
| *Family dynamics and structure* | | | | | | | | | |
| **Socializing with siblings** | | | | | | | | | |
| No | 110 (27.8) | 27.5 | 32.3 | 1.00 | 0.377 | 1.00 | 0.404 | – | – |
| Yes | 285 (72.2) | 72.5 | 67.7 | 0.80 [0.48-1.33] | | 0.78 [0.44-1.41] | | – | |
| **Parents or guardians work** | | | | | | | | | |
| No | 41 (13.6) | 13.3 | 14.8 | 1.00 | 0.734 | 1.00 | 0.609 | – | – |
| Yes | 260 (86.4) | 86.7 | 85.2 | 0.88 [0.43-1.90] | | 0.82 [0.38-1.81] | | – | |
| **Parents or guardians are in a relationship** | | | | | | | | | |
| No | 153 (42.1) | 37.7 | 55.3 | 1.00 | **0.003** | 1.00 | **<0.001** | 1.00 | **0.023** |
| Yes | 210 (57.9) | 62.3 | 44.7 | 0.49 [0.30-0.79] | | 0.33 [0.18-0.62] | | 0.38 [0.15-0.87] | |
| **Household headed exclusively by a woman** | | | | | | | | | |
| No | 296 (74.9) | 75.7 | 73.7 | 1.00 | 0.702 | 1.00 | 0.055 | – | – |
| Yes | 99 (25.1) | 24.3 | 26.3 | 1.11 [0.64-1.88] | | 0.48 [0.22-1.02] | | – | |
| *Adversities at home* | | | | | | | | | |
| **Perceived increase in the volume of domestic tasks** | | | | | | | | | |
| No | 256 (71.9) | 74.1 | 69.5 | 1.00 | 0.398 | 1.00 | 0.576 | – | – |
| Yes | 100 (28.1) | 25.9 | 30.5 | 1.26 [0.74-2.12] | | 0.83 [0.42-1.59] | | – | |
| **Lacked sufficient financial resources to procure adequate nourishment[1]** | | | | | | | | | |
| No | 284 (78.5) | 79.9 | 76.3 | 1.00 | 0.480 | 1.00 | 0.510 | – | – |
| Yes | 78 (21.5) | 20.1 | 23.7 | 1.23 [0.68-2.18] | | 0.78 [0.36-1.62] | | – | |

*(Continued)*

| Selected variables | n (%) | Moderate to severe level of anxiety (GAD-7) | | OR [95%CI] | p-value | Binary Logistic Models | | | |
|---|---|---|---|---|---|---|---|---|---|
| | | No | Yes | | | Model 1* | | Model 2** | |
| | | % | % | | | OR$_{adj}$ [95%CI] | p-value | OR$_{adj}$ [95%CI] | p-value |
| **Feeling afraid or very bad about being insulted/ rejected by adults at home** | | | | | | | | | |
| No | 229 (65.4) | 77.9 | 34.4 | 1.00 | **<0.001** | 1.00 | **0.001** | 1.00 | **0.010** |
| Yes | 121 (34.6) | 22.1 | 65.6 | 6.70 [3.96-11.5] | | 3.04 [1.53-6.01] | | 3.79 [1.36-10.97] | |
| **Feeling of lack of love and care from people at home** | | | | | | | | | |
| No | 223 (62.3) | 74.0 | 33.3 | 1.00 | **<0.001** | 1.00 | **<0.001** | 1.00 | **0.031** |
| Yes | 135 (37.7) | 26.0 | 66.7 | 5.70 [3.42-9.70] | | 3.73 [1.85-7.60] | | 3.06 [1.10-8.63] | |
| **Witnessing/experiencing situations of aggression, threats, or beatings at home** | | | | | | | | | |
| No | 322 (88.0) | 90.3 | 83.0 | 1.00 | 0.071 | 1.00 | 0.943 | 1.00 | 0.748 |
| Yes | 44 (12.0) | 9.7 | 17.0 | 1.91 [0.94-3.78] | | 1.03 [0.42-2.54] | | 1.25 [0.30-5.01] | |
| *Peer sociability and leisure activities* | | | | | | | | | |
| **Spent time hanging out with friends in person** | | | | | | | | | |
| No | 244 (68.7) | 68.1 | 67.0 | 1.00 | 0.849 | 1.00 | 0.779 | – | – |
| Yes | 111 (31.3) | 31.9 | 33.0 | 1.05 [0.63-1.74] | | 1.09 [0.58-2.02] | | – | |
| **Reached out to friends via messages or social networks** | | | | | | | | | |
| No | 74 (20.6) | 19.6 | 21.9 | 1.00 | 0.638 | 1.00 | 0.590 | – | – |
| Yes | 286 (79.4) | 80.4 | 78.1 | 0.87 [0.49-1.58] | | 0.80 [0.35-1.86] | | – | |
| **Chatted with friends via messages or social networks** | | | | | | | | | |
| No | 148 (41.3) | 42.4 | 37.9 | 1.00 | 0.448 | 1.00 | 0.753 | – | – |
| Yes | 210 (58.7) | 57.6 | 62.1 | 1.21 [0.74-1.98] | | 1.11 [0.58-2.12] | | – | |
| **Used social networks to chat with friends or interacted with computer games or other media[2]** | | | | | | | | | |
| No | 162 (48.6) | 56.9 | 33.7 | 1.00 | **<0.001** | 1.00 | **0.001** | 1.00 | **<0.001** |
| Yes | 171 (51.4) | 43.1 | 66.3 | 2.59 [1.55-4.41] | | 2.56 [1.44-4.67] | | 5.08 [2.08-13.52] | |
| **TV use[2]** | | | | | | | | | |
| No | 263 (76.9) | 80.7 | 73.0 | 1.00 | 0.141 | 1.00 | 0.396 | – | – |
| Yes | 79 (23.1) | 19.3 | 27.0 | 1.55 [0.86-2.73] | | 1.33 [0.68-2.52] | | – | |
| **Perceived improvement in the amount of contact with friends during the pandemic** | | | | | | | | | |
| No | 187 (55.5) | 54.3 | 55.1 | 1.00 | 0.908 | 1.00 | 0.862 | – | – |
| Yes | 150 (44.5) | 45.7 | 44.9 | 0.97 [0.59-1.59] | | 0.94 [0.45-1.95] | | – | |
| **Perceived improvement in the quality of relationships with friends during the pandemic** | | | | | | | | | |
| No | 166 (48.7) | 50.0 | 49.4 | 1.00 | 0.927 | 1.00 | 0.611 | – | – |
| Yes | 175 (51.3) | 50.0 | 50.6 | 1.02 [0.62-1.68] | | 1.20 [0.59-2.43] | | – | |
| *School setting* | | | | | | | | | |
| **Level of schooling** | | | | | | | | | |
| Elementary school | 277 (72.5) | 75.5 | 63.9 | 1.00 | **0.033** | 1.00 | **0.037** | 1.00 | **0.017** |
| High school | 105 (27.5) | 24.5 | 36.1 | 1.74 [1.04-2.88] | | 1.87 [1.04-3.36] | | 3.19 [1.21-8.89] | |
| **Level of schooling to be achieved** | | | | | | | | | |
| High school | 85 (27.8) | 27.0 | 22.5 | 1.00 | 0.432 | 1.00 | 0.926 | 1.00 | **0.042** |
| Higher education | 221 (72.2) | 73.0 | 77.5 | 1.27 [0.70-2.39] | | 1.03 [0.54-2.03] | | 0.34 [0.12-0.96] | |

*(Continued)*

**Table 2.** (Continued)

| Selected variables | n (%) | Moderate to severe level of anxiety (GAD-7) | | OR [95%CI] | p-value | Binary Logistic Models | | | |
|---|---|---|---|---|---|---|---|---|---|
| | | No | Yes | | | Model 1* | | Model 2** | |
| | | % | % | | | OR$_{adj}$ [95%CI] | p-value | OR$_{adj}$ [95%CI] | p-value |
| **Feeling cared for/protected at school by other adults** | | | | | | | | | |
| No | 37 (11.4) | 6.9 | 19.8 | 1.00 | **0.002** | 1.00 | **0.031** | 1.00 | 0.571 |
| Yes | 288 (88.6) | 93.1 | 80.2 | 0.30 [0.14-0.65] | | 0.39 [0.16-0.92] | | 0.68 [0.17-2.59] | |
| *Pandemic hurdles in education* | | | | | | | | | |
| **The pandemic has had a negative effect on the ability to focus on classes** | | | | | | | | | |
| No | 221 (65.2) | 69.1 | 54.0 | 1.00 | **0.013** | 1.00 | 0.326 | – | – |
| Yes | 118 (34.8) | 30.9 | 46.0 | 1.90 [1.14-3.16] | | 1.37 [0.73-2.54] | | – | |
| **The pandemic has had a negative effect on the ability to get good grades at school** | | | | | | | | | |
| No | 246 (72.8) | 76.6 | 60.7 | 1.00 | **0.005** | 1.00 | **0.038** | 1.00 | **0.050** |
| Yes | 92 (27.2) | 23.4 | 39.3 | 2.12 [1.25-3.59] | | 2.00 [1.04-3.88] | | 2.24 [1.02-5.49] | |
| **Considered dropping out of school** | | | | | | | | | |
| No | 335 (89.3) | 90.4 | 85.4 | 1.00 | 0.196 | 1.00 | 0.679 | – | – |
| Yes | 40 (10.7) | 9.6 | 14.6 | 1.61 [0.77-3.25] | | 1.19 [0.50-2.69] | | – | |

OR: Odds ratio. OR$_{adj}$: Adjusted odds ratio. 95%CI: 95% confidence interval.

[1]Referring to the preceding month when the survey was carried out.

[2]Use of more than 3 hours/day.

*Model 1: Adjusted within each domain, controlling for covariates specific to that domain.

**Model 2: Adjusted across domains, including variables with p < 0.05 in Model 1, as well as confounders ("Age", "Witnessing/experiencing situations of aggression, threats, or beatings at home", and "Level of schooling to be achieved"). Hosmer-Lemeshow goodness-of-fit test (p = 0.605); Pseudo-R$^2$ (Nagelkerke) = 0.833.

There was no evidence of multicollinearity (all VIF values were less than 5).

Values in bold indicate statistical significance (p < 0.05).

households, exposure to intrafamilial violence (including aggression, threats, or beatings), social interaction with peers (both in-person and online), and educational aspirations (Table 3).

## Discussion

This study addressed the challenging contexts faced by Brazilian adolescents residing in impoverished communities of São Paulo, regions that are typified by elevated societal vulnerability. These settings had immediate repercussions on the mental health of these adolescents, as evidenced by the results. In total, over two-thirds of the participants were classified as experiencing a high to moderate level of perceived stress, while just over a quarter demonstrated a moderate to severe level of anxiety. Among these, a higher proportion were girls.

The levels of anxiety observed in this study are consistent with those identified in the study by Sabbagh et al. (2022) [14], which was conducted in 25 countries and informed higher levels of anxiety among girls. The research by Halldorsdottir et al. (2021) [7] yielded similar results in Iceland. In France, Bourion-Bédès et al. (2024) [35] reported that adolescents experiencing domestic conflicts coupled with social isolation exhibited elevated levels of anxiety. Regarding the stress levels experienced by the participants in this study, 72.5% of adolescents were classified as exhibiting moderate to high stress, with a higher incidence among girls. Our findings align with the results of the systematic review by Iglesia

**Table 3.** Selected variables and associations with moderate to high level of perceived stress, as measured by PSS-10. São Paulo, 2021.

| Selected variables | n (%) | Moderate to high level of perceived stress (PSS-10) | | OR [95%CI] | p-value | Binary Logistic Models | | | |
|---|---|---|---|---|---|---|---|---|---|
| | | No | Yes | | | Model 1* | | Model 2** | |
| | | % | % | | | OR$_{adj}$ [95%CI] | p-value | OR$_{adj}$ [95%CI] | p-value |
| *Sociodemographic characteristics* | | | | | | | | | |
| **Gender** | | | | | | | | | |
| Girls | 203 (51.3) | 31.9 | 58.3 | 1.00 | **<0.001** | 1.00 | **<0.001** | 1.00 | **0.001** |
| Boys | 193 (48.7) | 68.1 | 41.7 | 0.33 [0.20-0.55] | | 0.33 [0.19-0.58] | | 0.28 [0.12-0.62] | |
| **Age** | | | | | | | | | |
| < 14 | 117 (29.5) | 37.4 | 23.7 | 1.00 | **0.015** | 1.00 | **0.039** | 1.00 | **0.031** |
| ≥ 14 | 279 (70.5) | 62.2 | 76.2 | 1.92 [1.14-3.21] | | 1.86 [1.03-3.35] | | 2.43 [1.08-5.56] | |
| **Race/skin color** | | | | | | | | | |
| Non-Black | 146 (36.9) | 39.6 | 34.2 | 1.00 | 0.362 | 1.00 | 0.348 | – | – |
| Black | 250 (63.1) | 60.4 | 65.8 | 1.26 [0.76-2.07] | | 1.33 [0.73-2.38] | | – | |
| **With a religious affiliation** | | | | | | | | | |
| No | 105 (26.5) | 20.9 | 31.7 | 1.00 | **0.048** | 1.00 | 0.424 | – | – |
| Yes | 291 (73.5) | 79.1 | 68.3 | 0.57 [0.31-1.00] | | 0.76 [0.38-1.47] | | – | |
| **School's administrative domain** | | | | | | | | | |
| Public | 341 (90.7) | 88.5 | 90.6 | 1.00 | 0.590 | 1.00 | 0.760 | 1.00 | 0.078 |
| Private | 35 (9.3) | 11.5 | 9.4 | 0.80 [0.37-1.84] | | 1.15 [0.48-2.85] | | 2.66 [0.90-8.62] | |
| **Owned housing** | | | | | | | | | |
| No | 150 (39.3) | 28.4 | 43.2 | 1.00 | **0.014** | 1.00 | **0.008** | 1.00 | **0.001** |
| Yes | 232 (60.7) | 71.6 | 56.8 | 0.52 [0.30-0.88] | | 0.46 [0.25-0.83] | | 0.29 [0.13-0.64] | |
| **Mother's schooling** | | | | | | | | | |
| Up to high school diploma or technical degree | 287 (82.2) | 79.5 | 81.0 | 1.00 | 0.780 | 1.00 | 0.827 | – | – |
| Higher education or above (incomplete or complete) | 62 (17.8) | 20.5 | 19.0 | 0.92 [0.50-1.74] | | 1.08 [0.54-2.21] | | – | |
| *Family dynamics and structure* | | | | | | | | | |
| **Socializing with siblings** | | | | | | | | | |
| No | 110 (27.8) | 31.9 | 25.4 | 1.00 | 0.243 | 1.00 | 0.451 | – | – |
| Yes | 285 (72.2) | 68.1 | 74.6 | 1.37 [0.80-2.32] | | 1.26 [0.68-2.30] | | – | |
| **Parents or guardians work** | | | | | | | | | |
| No | 41 (13.6) | 13.3 | 12.6 | 1.00 | 0.878 | 1.00 | 0.949 | – | – |
| Yes | 260 (86.4) | 86.7 | 87.4 | 1.06 [0.46-2.29] | | 0.97 [0.40-2.18] | | – | |
| **Parents or guardians are in a relationship** | | | | | | | | | |
| No | 153 (42.1) | 37.6 | 45.7 | 1.00 | 0.201 | 1.00 | 0.1211 | – | – |
| Yes | 210 (57.9) | 62.4 | 54.3 | 0.72 [0.43-1.19] | | 0.58 [0.28-1.15] | | – | |
| **Household headed exclusively by a woman** | | | | | | | | | |
| No | 296 (74.9) | 78.0 | 74.6 | 1.00 | 0.512 | 1.00 | 0.834 | 1.00 | 0.428 |
| Yes | 99 (25.1) | 22.0 | 25.4 | 1.21 [0.69-2.19] | | 0.91 [0.39-2.14] | | 1.42 [0.60-3.48] | |
| *Adversities at home* | | | | | | | | | |
| **Perceived increase in the volume of domestic tasks** | | | | | | | | | |
| No | 256 (71.9) | 78.4 | 70.1 | 1.00 | 0.135 | 1.00 | 0.535 | – | – |
| Yes | 100 (28.1) | 21.6 | 29.9 | 1.55 [0.88-2.83] | | 1.24 [0.64-2.46] | | – | |
| **Lacked sufficient financial resources to procure adequate nourishment[1]** | | | | | | | | | |
| No | 284 (78.5) | 86.4 | 77.1 | 1.00 | 0.059 | 1.00 | 0.321 | – | – |
| Yes | 78 (21.5) | 13.6 | 22.9 | 1.88 [0.98-3.89] | | 1.47 [0.69-3.32] | | – | |

*(Continued)*

**Table 3.** (Continued)

| Selected variables | n (%) | Moderate to high level of perceived stress (PSS-10) | | OR [95%CI] | p-value | Binary Logistic Models | | | |
|---|---|---|---|---|---|---|---|---|---|
| | | No | Yes | | | Model 1* | | Model 2** | |
| | | % | % | | | OR$_{adj}$ [95%CI] | p-value | OR$_{adj}$ [95%CI] | p-value |
| **Feeling afraid or very bad about being insulted/ rejected by adults at home** | | | | | | | | | |
| No | 229 (65.4) | 88.6 | 57.2 | 1.00 | **<0.001** | 1.00 | **0.019** | 1.00 | 0.181 |
| Yes | 121 (34.6) | 11.4 | 42.8 | 5.83 [2.99-12.5] | | 2.61 [1.16-6.24] | | 1.99 [0.73-5.80] | |
| **Feeling of lack of love and care from people at home** | | | | | | | | | |
| No | 223 (62.3) | 89.0 | 51.8 | 1.00 | **<0.001** | 1.00 | **<0.001** | 1.00 | **0.002** |
| Yes | 135 (37.7) | 11.0 | 48.2 | 7.54 [3.87-16.2] | | 4.00 [1.73-10.2] | | 4.80 [1.71-15.0] | |
| **Witnessing/experiencing situations of aggression, threats, or beatings at home** | | | | | | | | | |
| No | 322 (88.0) | 94.5 | 85.7 | 1.00 | **0.019** | 1.00 | 0.593 | 1.00 | 0.727 |
| Yes | 44 (12.0) | 5.5 | 14.3 | 2.87 [1.17-8.61] | | 0.72 [0.22-2.53] | | 0.76 [0.17-3.72] | |
| *Peer sociability and leisure activities* | | | | | | | | | |
| **Spent time hanging out with friends in person** | | | | | | | | | |
| No | 244 (68.7) | 64.0 | 68.6 | 1.00 | 0.443 | 1.00 | 0.811 | 1.00 | 0.460 |
| Yes | 111 (31.3) | 36.0 | 31.4 | 0.82 [0.49-1.38] | | 1.08 [0.59-2.00] | | 0.74 [0.33-1.66] | |
| **Reached out to friends via messages or social networks** | | | | | | | | | |
| No | 74 (20.6) | 16.7 | 20.5 | 1.00 | 0.427 | 1.00 | 0.768 | – | – |
| Yes | 286 (79.4) | 83.3 | 79.5 | 0.77 [0.40-1.44] | | 1.13 [0.48-2.57] | | – | |
| **Chatted with friends via messages or social networks** | | | | | | | | | |
| No | 148 (41.3) | 33.3 | 44.2 | 1.00 | 0.073 | 1.00 | 0.056 | 1.00 | **0.015** |
| Yes | 210 (58.7) | 66.7 | 55.8 | 0.63 [0.37-1.04] | | 0.53 [0.27-1.02] | | 0.39 [0.18-0.84] | |
| **Used social networks to chat with friends or interacted with computer games or other media[2]** | | | | | | | | | |
| No | 162 (48.6) | 56.1 | 48.2 | 1.00 | 0.220 | 1.00 | 0.291 | – | – |
| Yes | 171 (51.4) | 43.9 | 51.8 | 1.38 [0.83-2.30] | | 1.36 [0.77-2.41] | | – | |
| **TV use[2]** | | | | | | | | | |
| No | 263 (76.9) | 84.5 | 75.9 | 1.00 | 0.094 | 1.00 | 0.130 | – | – |
| Yes | 79 (23.1) | 15.5 | 24.1 | 1.73 [0.91-3.50] | | 1.70 [0.86-3.55] | | – | |
| **Perceived improvement in the amount of contact with friends during the pandemic** | | | | | | | | | |
| No | 187 (55.5) | 53.7 | 57.6 | 1.00 | 0.094 | 1.00 | 0.343 | – | – |
| Yes | 150 (44.5) | 46.3 | 42.4 | 0.85 [0.51-1.42] | | 1.40 [0.70-2.85] | | – | |
| **Perceived improvement in the quality of relationships with friends during the pandemic** | | | | | | | | | |
| No | 166 (48.7) | 41.7 | 54.1 | 1.00 | 0.051 | 1.00 | 0.132 | – | – |
| Yes | 175 (51.3) | 58.3 | 45.9 | 0.61 [0.36-1.00] | | 0.59 [0.29-1.17] | | – | |
| *School setting* | | | | | | | | | |
| **Level of schooling** | | | | | | | | | |
| Elementary school | 277 (72.5) | 75.0 | 70.6 | 1.00 | 0.428 | 1.00 | 0.332 | – | – |
| High school | 105 (27.5) | 25.0 | 29.4 | 1.25 [0.72-2.22] | | 1.36 [0.73-2.62] | | – | |

*(Continued)*

**Table 3.** (Continued)

| Selected variables | n (%) | Moderate to high level of perceived stress (PSS-10) | | OR [95%CI] | p-value | Binary Logistic Models | | | |
|---|---|---|---|---|---|---|---|---|---|
| | | No | Yes | | | Model 1* | | Model 2** | |
| | | % | % | | | OR$_{adj}$ [95%CI] | p-value | OR$_{adj}$ [95%CI] | p-value |
| **Level of schooling to be achieved** | | | | | | | | | |
| High school | 85 (27.8) | 21.8 | 29.3 | 1.00 | 0.201 | 1.00 | 0.081 | 1.00 | 0.033 |
| Higher education | 221 (72.2) | 78.2 | 70.7 | 0.67 [0.35-1.23] | | 0.54 [0.26-1.08] | | 0.39 [0.15-0.93] | |
| **Feeling cared for/protected at school by other adults** | | | | | | | | | |
| No | 37 (11.4) | 2.7 | 12.8 | 1.00 | **0.005** | 1.00 | **0.016** | 1.00 | **0.050** |
| Yes | 288 (88.6) | 97.3 | 87.2 | 0.19 [0.03-0.65] | | 0.22 [0.03-0.78] | | 0.18 [0.01-0.98] | |
| *Pandemic hurdles in education* | | | | | | | | | |
| **The pandemic has had a negative effect on the ability to focus on classes** | | | | | | | | | |
| No | 221 (65.2) | 72.2 | 64.3 | 1.00 | 0.197 | 1.00 | 0.903 | – | – |
| Yes | 118 (34.8) | 27.8 | 35.7 | 1.44 [0.83-2.57] | | 1.04 [0.54-2.03] | | – | |
| **The pandemic has had a negative effect on the ability to get good grades at school** | | | | | | | | | |
| No | 246 (72.8) | 80.0 | 70.5 | 1.00 | 0.095 | 1.00 | 0.087 | – | – |
| Yes | 92 (27.2) | 20.0 | 29.5 | 1.67 [0.92-3.19] | | 1.90 [0.91-4.15] | | – | |
| **Considered dropping out of school** | | | | | | | | | |
| No | 335 (89.3) | 96.5 | 87.6 | 1.00 | **0.009** | 1.00 | **0.020** | – | – |
| Yes | 40 (10.7) | 3.5 | 12.4 | 3.93 [1.35-16.7] | | 3.55 [1.19-15.3] | | – | |

OR: Odds ratio. OR$_{adj}$: Adjusted odds ratio. 95%CI: 95% confidence interval.

[1]Referring to the preceding month when the survey was carried out.

[2]Use of more than 3 hours/day.

*Model 1: Adjusted within each domain, controlling for covariates specific to that domain.

**Model 2: Adjusted across domains, including variables with $p < 0.05$ in Model 1, as well as confounders ("School's administrative domain", "Household headed exclusively by a woman", "Witnessing/experiencing situations of aggression, threats, or beatings at home", "Spent time hanging out with friends in person", "Chatted with friends via messages or social networks" and "Level of schooling to be achieved"). Hosmer-Lemeshow goodness-of-fit test ($p = 0.753$); Pseudo-$R^2$ (Nagelkerke) = 0.407.

The variable "Considered dropping out of school" was removed from the adjusted model due to evidence of multicollinearity (VIF ≥ 5).

Values in bold indicate statistical significance ($p < 0.05$).

and Lu (2021) [13], which emphasized the necessity for institutional support to mitigate the psychological impacts of the pandemic.

The advent of the Covid-19 pandemic has precipitated substantial alterations in the social interactions, employment landscape, educational sector, and familial structures [6]. In Brazil, the most impoverished families were particularly impaired, as their primary source of income was rapidly diminishing, necessitating reductions in essential expenditures for human survival, including food, medicine, and housing [36]. The circumstances encountered by most of the young people in this study also substantiate this assertion, as those who did not own their homes were compelled to confront this form of insecurity and uncertainty.

The public health crisis was further exacerbated by the Executive Branch's denial of scientific evidence and devaluation of public bodies during this period [37]. This manifested itself in the choice and spread of unproven treatments and the trivialization of the seriousness of the pandemic, to the detriment of an approach that prioritized collective

wholesomeness, especially of the most vulnerable groups [38]. These groups were deprived of any form of State support, which has had a detrimental impact on their mental health.

In response to the dearth of efficacious treatments, social isolation emerged as a dominant coping mechanism in Brazil and numerous other regions worldwide [39]. The closure of public institutions, including schools, places of worship, and commercial establishments, resulted in a significant social and economic restructuring [40]. This measure had disparate impacts on different population segments, varying in degree and across multiple dimensions as the crisis intensified [3,41]. Women were disproportionately affected [42], and socially marginalized groups facing economic hardship, racial discrimination, or gender nonconformity also reported worse outcomes [9,43].

Our findings indicated no higher levels of anxiety or perceived stress in situations where siblings lived together, where the parents' or guardians' employment status remained stable, or where the household was headed exclusively by women, thereby challenging the assertions made by the mainstream media and some scientific literature [44]. Moreover, the presence of parents in a conjugal relationship emerged as a significant protective factor against moderate to severe levels of anxiety, suggesting that the continuity of parental bonds may serve as a stabilizing element amid broader routine disruptions. Nonetheless, the intensification of domestic coexistence resulted from school closures and diminished contact with peers and external social networks reshaped familial dynamics in ways not necessarily marked by effective communication or mutual understanding between caregivers and children [8].

While the amount of time spent at home increased for both boys and girls, the impact on mental health stood apart significantly according to gender. For girls, the home environment frequently constituted a source of anxiety and stress, as they more frequently reported feelings of fear/rejection, and the absence of love and care from adults. This phenomenon reveals the detrimental impact of adverse familial dynamics on psychological well-being. Conversely, although boys also remained more time at home than usual during the pandemic, they reported fewer negative repercussions stemming from family relationships. For many of them, it was still feasible to cultivate and maintain bonds with peers, who functioned as an important source of emotional regulation and a protective buffer against adversity. This dynamic appears closely linked to gendered asymmetries in autonomy and circulation: data from the "For being a Girl in Brazil" research (2021) [45] revealed that 54% of girls experienced an increased domestic workload during the pandemic, which significantly limited their possibilities for socialization and reinforced their confinement within the household sphere. Meanwhile, boys were more often permitted – and even expected – to engage in activities outside the home, such as informal work, errands, or street-level leisure, which facilitated greater access to relational networks and to spaces of recognition and escape.

Such distinctions are embedded in broader patriarchal logics that naturalize the idea of the home as a legitimate and obligatory place for girls, where they are expected to embody care, docility, and discretion [17,18]. For boys, in contrast, mobility and confrontation, whether with family norms or external situations, are symbolically and socially tolerated, if not encouraged. In this configuration, while the home may have represented a site of psychological fragility for girls, particularly due to overburden and surveillance, for boys it was possible to negotiate its tensions by mobilizing external supports [15]. This interpretation aligns with the extant literature indicating that the quality and availability of social interactions during the pandemic acted as a moderating factor for the effects of intrafamilial stress, particularly among adolescents with broader mobility and peer access, thus mitigating adverse mental health outcomes [46].

Another point that warrants careful consideration is the issue of race/skin color. Although this variable did not show a statistically significant association with the mental health outcomes in our adjusted models, the fact that more than 60% of the adolescent respondents self-identified as Black requests a deeper analytical reflection. Such demographic pattern is not incidental, but indicative of a broader processes of racialized inequality that shape the Brazilian urban periphery. As emphasized by Moura et al. (2024) [47], Black and mixed-race populations residing in socioeconomically vulnerable territories were disproportionately affected by the repercussions of the Covid-19 pandemic, particularly in terms of food insecurity and income instability, while white individuals were comparatively less exposed to such vulnerabilities.

Additionally, Santos et al. (2023) [48] argued that the pandemic exacerbated the historical effects of structural racism, revealing how race continues to shape living conditions, access to healthcare, and exposure to psychosocial stressors. In this light, the lack of statistical significance in our findings should not be interpreted as a sign of irrelevance; rather, it reflects the limitations of a unidimensional quantitative interpretation in grasping the complex structural, symbolic, and institutional dynamics through which race functions as both a determinant of vulnerability and a force that shapes how adolescent mental health is experienced, expressed, and made visible.

Another variable of interest was age, which showed a positive association with perceived stress, aligning with findings from van Loon et al. (2022) [49], who observed a similar trend among Dutch adolescents during the Covid-19 pandemic. The study suggests that older adolescents may be more susceptible to stress due to the cumulative demands of academic transitions, identity development, and shifting social dynamics. In this context, academic performance constituted a salient anxiogenic factor, as difficulties in maintaining satisfactory grades during this turbulent period were frequently reported as sources of psychological distress. Nonetheless, the perception of school as a protective space appeared to mitigate stress levels among participants. These findings suggest that, despite the educational setbacks, schools continued to play a crucial role as relational spaces, where adolescents could build supportive bonds and access trusted adult figures, highlighting the importance of interpersonal connections within the school context [50,51].

The prolonged closure of schools during the pandemic significantly impacted adolescents' well-being, contributing to heightened anxiety and perceived stress. As key spaces of security and support, the absence of schools likely intensified feelings of distress and uncertainty about the future among many young people [8]. Within these considerations, educational and public health policies are of paramount importance in the lives of adolescents. In the event of a future health emergency, it is of the utmost importance that strategies be developed to mitigate the side effects of school closures, including the reinforcement of virtual support networks, the training of educators in the identification of signs of psychosocial distress, and the assurance that the return to face-to-face classes is conducted in a secure and well-planned manner. Such measures will assist in restoring the protective environment that the school represents.

The differentiation between behaviors that promote well-being and those that precipitate stress and anxiety is frequently tenuous within the milieu of digital interactions. During the pandemic, the increase in time spent online has been both a source of relief and a source of stress for many adolescents [52]. Although digital technologies provide a conduit for social interaction, excessive time spent on these activities might exacerbate anxiety among adolescents. This is due to several factors, including heightened social comparisons, constant exposure to an overload of negative information, an increased risk of feelings of isolation, and the nature of the content consumed. Albeit these factors have been identified as having a detrimental impact on the mental health of young people, they also served as a means of maintaining contact with their peers during the pandemic [8]. Consequently, they functioned as a form of stress relief, particularly in the context of navigating health-related challenges [53].

Housing ownership emerged as a protective factor against perceived stress, supporting the idea that residential stability contributes to emotional well-being. As suggested by Cosma et al. (2023) [46], secure living conditions may buffer adolescents from chronic stress by reducing daily uncertainties and reinforcing a sense of safety and control over their environment. It is therefore imperative that strategies to promote family and social support be implemented without delay, especially in contexts of socio-economic vulnerability. Public health and education policies ought to integrate these protective dimensions into their strategies for addressing the psychosocial challenges that will undoubtedly emerge in the context of forthcoming health emergencies.

This study is not without limitations. First, the absence of data on the context prior to the pandemic precludes comparisons with stressful or anxiety-inducing factors that may have been present in the adolescents' lives [32]. Furthermore, the non-probabilistic sample does not allow for the results to be generalized, despite providing unprecedented data on the mental health of boys and girls amid the pandemic in peripheral settings in Brazil. Third, the timing of data collection likely attenuated immediate pandemic-related stressors; however, it captured a critical moment of psychological readjustment,

shaped by both recovery dynamics and persistent vulnerabilities. A final point pertains to the scales' construction and validation, rather than to the study directly; the observed associations between gender/anxiety and gender/perceived stress (in which boys consistently appear more protected) may also be indicative of a distinctive feature of the gender-based norms to which both boys and girls are subjected.

Based on these considerations, it appears prudent to put forth the following hypothesis: gender norms tend to encourage girls to display markedly more emotional expressiveness from an early age, while boys are socialized to suppress their emotional vulnerabilities, aligning (or adjusting) themselves to standards of masculinity that prioritize traits such as virility, strength, and restraint in displaying affections. Moreover, these norms generally associate emotional vulnerability with weakness.

The role of machismo and gender-based stereotypes in how boys cope with their emotions has been well documented [54]. In the broader context, the prescribed ideals concerning such conduct, which fail to acknowledge the diversity of masculinities, result in men repressing their emotions and feelings, even from their earliest development as social beings. This is exemplified by the popular adage "man don't cry", which contributes to an underestimation of hardships relating to mental health issues. Such a departure from the established behavioral norms challenges the conventional notions of masculinity and prompts a re-evaluation of the traditional dichotomy between "being a man" and "being a woman". It can therefore be posited that the scales may not only be measuring situations of greater vulnerability and emotional stress, especially for girls, but they may also be making explicit the effects of gender-based norms on the answers given (or rather, on the possibility of stating certain answers positively or negatively).

Gender norms exert distinct pressures on girls, particularly in patriarchal societies like the Brazilian one. While their emotional expression is not necessarily suppressed, it is subject to constant scrutiny and moral regulation. From an early age, girls experience intensified domestic oversight, facing stricter controls over their behavior, mobility, and forms of self-expression. This surveillance, far from being a mere cultural artifact, functions as a mechanism of regulation with profound implications for mental health. When intertwined with symbolic and psychological violence within the family sphere, these constraints generate chronic emotional tension and a persistent sense of vulnerability [15,54].

International evidence consistently demonstrates a marked increase in domestic and gender-based violence during the Covid-19 pandemic, largely attributed to prolonged confinement, social isolation, and intensified stress within households [55–57]. In resonance with this global scenario, national research has revealed how the pandemic exacerbated gendered vulnerabilities [58]. Particularly, women residing in peripheral territories, who are predominantly Black, economically marginalized, and often solely responsible for sustaining their households, have endured a convergence of structural oppressions that intensified under pandemic conditions [59]. According to Campos et al. (2025) [60], almost 28% of women reported conjugal violence during this period, with significantly higher odds among those situated in contexts permeated by drug trafficking or precarious and informal labor arrangements. These data expose the relational and territorial dynamics of violence, in which structural abandonment and the presence of criminal economies operate in concert to deepen processes of dispossession and neglect. The prolongation of domestic confinement intensified not only the frequency of interpersonal aggression, but also the mechanisms of moral regulation, affective constraint, and economic precariousness that disproportionately shape the trajectories of girls and women [61].

Such experiences must be understood as expressions of enduring gendered logics that delineate the very contours through which suffering is codified, legitimized, and articulated. In moments of intensified crisis, such as the pandemic, the normative frameworks affect the visibility of emotional suffering by inscribing the permissible forms of its manifestation along deeply gendered lines.

## Conclusion

This study illuminated the array of defiant scenarios confronting adolescents residing in a peripheral region of São Paulo amid the Covid-19 pandemic. These experiences are consistent with those previously documented for adolescents and

young people in other national [43] and international mounts [25,44]. The study identified protective factors within the family, school, and peer socialization spaces that have been demonstrated to be beneficial in the context of the adverse effects of the pandemic.

The adolescent period represents a crucial phase during which the maturation of social and emotional competencies reaches its peak [5]. The primary agents of socialization during this period – namely, the family, school, and peer network – play a pivotal role in the process of consolidating skills and establishing a social identity [5]. The Covid-19 pandemic has had a profound impact on the daily lives of individuals across the globe, with deeper implications for those under the care of adults. The restrictions on movement, the intensification of domestic interactions (often described as conflictive or negative), the longer online connection, and the complete disruption of the academic routine introduced a plethora of stressful and anxiogenic factors for adolescents during this period.

Adverse effects were voiced with greater frequency by girls. We posit that this is the result of a multiplicity of factors related to gender-based experience and expression. While the disproportionate representation of female participants in contexts of social and relational vulnerability may partially account for this trend, it is equally crucial to consider that girls are not only more susceptible to psychosocial suffering but are also more inclined (or permitted within prevailing gender norms) to externalize such suffering through heightened emotional expressivity. This dual condition, of intensified vulnerability and socially sanctioned expressiveness, may help explain the lower scores observed among boys on measures of higher levels of anxiety and perceived stress when compared to their female counterparts.

It is not possible to determine the extent to which these results are attributable to the subjects' experiencing contexts that are, indeed, less anxiogenic or stressful for them, or whether their superior performance on the scales is due to a cultural context that values strength and the restraint of emotions. Notwithstanding, our findings indicate that girls have also been significantly affected by the Covid-19 pandemic. This lends further support to the argument that extraordinary health crises cannot be viewed as homogeneous phenomena. Rather, they are subject to significant variations according to several factors, inclusive of gender, race, social class, age, and other social markers of difference and inequality.

## Supporting information

**S1 Data. Coded dataset derived from participants' responses to the questionnaire survey.**
(XLSX)

## Author contributions

**Conceptualization:** Nicolas Kimura Generoso, Cristiane da Silva Cabral.

**Formal analysis:** Nicolas Kimura Generoso, Ana Luiza Vilela Borges.

**Funding acquisition:** Ana Luiza Vilela Borges.

**Investigation:** Nicolas Kimura Generoso, Cristiane da Silva Cabral.

**Methodology:** Nicolas Kimura Generoso, Ana Luiza Vilela Borges.

**Software:** Nicolas Kimura Generoso.

**Supervision:** Cristiane da Silva Cabral.

**Writing – original draft:** Nicolas Kimura Generoso, Cristiane da Silva Cabral, Ana Luiza Vilela Borges.

**Writing – review & editing:** Nicolas Kimura Generoso, Cristiane da Silva Cabral, Ana Luiza Vilela Borges.

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
