## [Decision Letter · Decision Letter 0]

30 May 2025

PMEN-D-25-00090

Gender differences in anxiety and perceived stress during the Covid-19 pandemic among Brazilian adolescents from impoverished communities

PLOS Mental Health

Dear Dr. Cabral,

Thank you for submitting your manuscript to PLOS Mental Health. After careful consideration, we feel that it has merit but does not fully meet PLOS Mental Health’s publication criteria as it currently stands. Therefore, we invite you to submit a revised version of the manuscript that addresses the points raised during the review process.

We look forward to receiving your revised manuscript.

Kind regards,

Dr. Carrie Anne Marshall

Academic Editor

PLOS Mental Health

Journal Requirements:

1. Please include a complete copy of PLOS’ questionnaire on inclusivity in global research in your revised manuscript. Our policy for research in this area aims to improve transparency in the reporting of research performed outside of researchers’ own country or community. The policy applies to researchers who have travelled to a different country to conduct research, research with Indigenous populations or their lands, and research on cultural artefacts. The questionnaire can also be requested at the journal’s discretion for any other submissions, even if these conditions are not met.  Please find more information on the policy and a link to download a blank copy of the questionnaire here: https://journals.plos.org/mentalhealth/s/best-practices-in-research-reporting. Please upload a completed version of your questionnaire as Supporting Information when you resubmit your manuscript.

2. In the online submission form, you indicated that [The data that support the findings of this study are available on request from the corresponding author, CSC.].

a. In a public repository,

b. Within the manuscript itself, or

c. Uploaded as supplementary information.

Additional Editor Comments (if provided):

Reviewers' comments:

Reviewer's Responses to Questions

**Comments to the Author**

1. Does this manuscript meet PLOS Mental Health’s publication criteria ? Is the manuscript technically sound, and do the data support the conclusions? The manuscript must describe methodologically and ethically rigorous research with conclusions that are appropriately drawn based on the data presented.

Reviewer #1: Yes

Reviewer #2: Partly

2. Has the statistical analysis been performed appropriately and rigorously?

Reviewer #1: Yes

Reviewer #2: Yes

3. Have the authors made all data underlying the findings in their manuscript fully available (please refer to the Data Availability Statement at the start of the manuscript PDF file)?

Reviewer #1: Yes

Reviewer #2: Yes

4. Is the manuscript presented in an intelligible fashion and written in standard English?

Reviewer #1: Yes

Reviewer #2: Yes

5. Review Comments to the Author

Reviewer #1: The manuscript is technically sound, and the data support the conclusions. The manuscript describes methodologically and ethically rigorous the topic addressed with studies that support their conclusionsconclusions that are appropriately drawn based on the data presented.

However, there is a point that I would like to discuss that refers to a hypothesis presented by the authors in light of the result that girls have greater experiences of anxiety and perceived stress than boys. According to the authors' hypothesis, the difference between genders refers to behavioral norms, impregnated by machismo, socially imposed, which make boys repress their emotional expressions, unlike girls who are encouraged to show their emotions more abundantly.

I do not disagree with this interpretation, but I believe that another interpretation is also possible. Women and girls, in general, in a society marked by patriarchy as an important social marker in Western society and in Brazil, a colonized society, receive a stricter education and have their behavior more controlled than men and boys. In other words, in a context of domestic confinement, the fear of suffering repression or insults from adults is much more frequent for girls than for boys, who are raised with much greater freedom of behavior. Girls are more vulnerable to domestic violence than boys. And the variable fear of being insulted or not receiving love from adults in the house may reflect situations of domestic violence, even if not physical, but possible to be psychologically applicable.

It is suggested that the authors also reflect on the issue with scientific literature that shows the rates of domestic violence and gender-based violence experienced during the COVID-19 pandemic.

Another point that I consider important is the issue related to race/skin color. Although the race/color factor did not show a statistically significant association with mental health experiences, the majority of the adolescent respondents were black (more than 60%). So some consideration I believe is important in the topic of discussion.

Reviewer #2: The study addresses an urgent and globally relevant issue the impact of COVID-19 on adolescent mental health, with a specific focus on Brazilian youth, adding valuable regional data to the global literature.

The background could benefit from a more focused review of existing studies on adolescent mental health during COVID-19, particularly in Latin America or similar socio-economic contexts, to better position the study’s contribution.

Regarding the sample, the use of a non-probabilistic sampling method limits generalizability, so it is important to clarify how participants were selected and to discuss potential biases. Providing details about demographic characteristics, such as socioeconomic status and ethnicity, would also help contextualize the findings. Data collection took place between November and December 2021, after the initial peak of the pandemic; therefore, it would be useful to address how this timing may have influenced the mental health assessments compared to earlier phases.

While the scales used in the study are well-described, the justification for the cutoff points, such as using GAD-7 ≥10 for moderate/severe anxiety, should be explicitly referenced, especially considering cultural validation. The statistical analysis section currently lacks a complete description of the multivariate modelling process, so the full methodology including criteria for variable inclusion or exclusion, handling of confounders, and checks for multicollinearity should be clearly detailed, and it would strengthen the paper to include model fit indices or measures of predictive validity.

6. PLOS authors have the option to publish the peer review history of their article (what does this mean? ). If published, this will include your full peer review and any attached files.

**Do you want your identity to be public for this peer review?** For information about this choice, including consent withdrawal, please see our Privacy Policy .

Reviewer #1: **Yes: ** Regina Celia Fiorati

Reviewer #2: No

---

## [Editor Report · Decision Letter 1]

31 Jul 2025

Gender differences in anxiety and perceived stress during the Covid-19 pandemic among Brazilian adolescents from impoverished communities

PMEN-D-25-00090R1

Dear Dr. Cabral,

We are pleased to inform you that your manuscript 'Gender differences in anxiety and perceived stress during the Covid-19 pandemic among Brazilian adolescents from impoverished communities' has been provisionally accepted for publication in PLOS Mental Health.

Best regards,

Carrie Marshall

Academic Editor

PLOS Mental Health